# Workplace Predictors of Violence against Nurses Using Machine Learning Techniques: A Cross-Sectional Study Utilizing the National Standard of Psychological Workplace Health and Safety

**DOI:** 10.3390/healthcare11071008

**Published:** 2023-04-01

**Authors:** Farinaz Havaei, Nassim Adhami, Xuyan Tang, Sheila A. Boamah, Megan Kaulius, Emili Gubskaya, Kenton O’Donnell

**Affiliations:** 1School of Nursing, The University of British Columbia, Vancouver, BC V6T 2B5, Canada; 2Department of Educational and Counselling Psychology, and Special Education, University of British Columbia, Vancouver, BC V6T 1Z4, Canada; 3School of Nursing, McMaster University, Hamilton, ON L8S 4K1, Canada; 4Faculty of Medicine, The University of British Columbia, Vancouver, BC V6T 1Z3, Canada

**Keywords:** nursing, workplace violence, machine learning

## Abstract

Background: Nurses experience an alarming rate of violence in the workplace. While previous work has indicated that working conditions play an important role in workplace violence outcomes, these studies have not used comprehensive and systematically operationalized variables. Methods: Through cross-sectional survey responses from 4066 British Columbian nurses, we identified which of the 13 psychosocial factors, as outlined in the National Standard of Psychological Workplace Health and Safety, are most predictive of workplace violence perpetrated against nurses by patients and their visitors (Type II violence) and organizational employees (Type III violence). Results: Eighty-seven percent of respondents indicated that they had experienced Type II violence, whereas 48% indicated they had experienced Type III violence over the last year. Lack of physical safety, workload management, and psychological protection were the top three psychosocial factors in the workplace predictive of Type II violence, whereas lack of civility and respect, organizational culture, and psychological support were the top three factors associated with Type III violence. Conclusions: The findings in this study shed light on the distinct psychosocial factors in the workplace in need of investment and intervention to address Type II and III violence.

## 1. Introduction

Nurses, worldwide, are at a high risk of exposure to workplace violence (WPV) [1,2,3,4,5], with some evidence indicating that nurses are four times more likely to be exposed to WPV compared to other healthcare workers [6]. Workplace violence (i.e., physical assault, threat of assault, emotional abuse, verbal sexual harassment, or sexual assault) is a costly phenomenon for any healthcare system, but even more so in a strained system experiencing significant human resource demand and inadequate supply [7]. Regardless of its various forms (e.g., physical or emotional) and sources (e.g., patients or co-workers), workplace violence results in negative outcomes for healthcare systems through increasing rates of worker psychological distress, absenteeism, turnover, and patient adverse events [3,7,8,9,10,11]. Previous work has indicated that exploring working conditions that are most predictive of workplace violence is an urgent need [12,13]. Furthermore, utilizing a comprehensive and systematically developed instrument to assess working conditions would allow for a greater understanding of potential predictors for workplace violence within and between sources.

Workplace violence is defined by the Canadian Federation of Nurses Union as “physical or non-physical violence which includes the threatened, attempted or actual work-related incident of physical force or psychological abuse which can result in physical, emotional and sexual injury, harm, or trauma” (para 1, 2023) [14]. Research often categorizes workplace violence according to its form (e.g., physical versus non-physical) and source (e.g., clients versus co-workers). An international review of 136 studies, inclusive of a sample of over 151,000 nurses worldwide, showed that non-physical workplace violence was the most common form of WPV, with 77% of nurses reporting that they had experienced non-physical violence at least once over the last year [4]. A more recent study with Canadian nurses classified workplace violence into physical assault, threat of assault, emotional abuse, verbal sexual harassment, and sexual assault [5]. This study consistently found non-physical violence, particularly emotional abuse (83%) and threat of assault (78%), was the most common form of workplace violence reported by nurses in Canada. These findings are concerning, particularly when considering evidence from a recent scoping review of nursing workplace violence that found that the reporting rate for incidents of workplace violence is often less than 50% [15]. 

Workplace violence is also classified according to its instigator source. The National Institute for Occupational Safety and Health in the US refers to workplace violence instigated by clients and consumers as Type II and those instigated by other organizational employees as Type III workplace violence [16]. An international review found patients (physical: 64%, non-physical: 54%) and patients’ family/friends (physical: 30%, non-physical: 47%), followed by co-workers, including nurses, physicians, and other staff members (physical: 1.6–6.3%, non-physical: 21.8–39.2%), were the most frequent instigators of workplace violence towards nurses [4]. Other research has revealed similar findings among Canadian nurses [5]. 

The implications of various forms of workplace violence in healthcare are well documented. A systematic review of 68 studies by Lanctot and Guay (2014) found exposure to both physical and non-physical violence had negative consequences for healthcare worker victims in relation to physical, psychological, emotional, performance, relational, social, and financial outcomes [7]. Psychological (e.g., post-traumatic stress), emotional (e.g., fear), and performance (e.g., absenteeism and job dissatisfaction) implications were perceived as the most severe and the most frequently reported consequences of healthcare worker exposure to workplace violence [7]. 

Despite these well-studied consequences, workplace predictors of violence towards nurses are less clearly understood. A systematic review of 13 studies examining the antecedents of Type II and III workplace violence towards nurses found that a variety of isolated working conditions were important predictors of WPV [13]. Workplace predictors of Type II violence included the absence of organizational safety/security measures, healthcare errors, the time of the day, working alone, unsupportive management, inadequate staffing, and work overload. Workplace predictors of Type III violence included factors such as staffing inadequacies, lack of professional training, a culture of normalization, lack of group cohesiveness, and lack of ineffective leadership. Despite this evidence, none of the individual studies included in the review used a comprehensive measure of working conditions to determine if certain psychosocial factors were more or less predictive of various types of workplace violence (i.e., most studies only included four to five workplace factors as predictors). As such, employing an instrument that is both systematically developed, and encompassing of a wide range of working conditions, would allow for a greater understanding of how specific psychosocial factors may influence workplace violence outcomes. 

One such option for an instrument was developed by the Mental Health Commission of Canada who used an extensive literature review, along with a grounded theory study, to systematically develop a comprehensive measure of working conditions important to the mental health and safety of workers in all sectors [17,18]. The findings of this work identified 13 psychosocial risk factors that were integrated into the National Standard of Workplace Psychological Health and Safety (“the Standard”). The Standard and its measurement tool were recently validated among Canadian nurses [19,20]. According to these studies, ineffective workload management and lack of or limited psychological protection in the workplace were consistently among the strongest predictors of five mental health and three care delivery outcomes. While the relationship between the Standard’s 13 psychosocial factors and mental health and care delivery outcomes has been demonstrated, the relationship between the Standard and workplace violence committed against nurses remains unclear. This study aimed to identify which of the 13 psychosocial risk factors, as outlined in the Standard, are most predictive of Type II and III workplace violence committed against nurses. 

This study was informed by theories of healthy work environments such as the nursing worklife model [21,22,23]. According to this model, healthy work environments have certain characteristics—strong leadership, adequate staffing and resources, plenty of opportunities for staff to participate in organizational affairs, collegial relations among staff, and a nursing model of care delivery—that result in positive outcomes for nurses (higher job satisfaction and lower burnout) [21,23] and patients (fewer adverse events) [22] and subsequently lower likelihood of aggression and violence from patients, their families, and the staff. 

## 2. Materials and Methods

### 2.1. Data Collection and Sample

Through a collaboration between nurse researchers and the British Columbia Nurses’ Union (BCNU), a cross-sectional correlational study was conducted in 2019. The BCNU, representing nearly 48,000 nurses within BC, invited its members to take part in the study via email with the survey link. Nurses were informed that their participation was anonymous and voluntary, and that submission of the survey would be regarded as informed consent. A total of 5512 nurses were recruited, yielding a response rate of approximately 12%. Nurses who were not actively working and did not specify their gender were removed, resulting in a total of 4066 participants. Nurses who did not report their gender accounted for a small proportion of the total sample (less than 2%); hence, we decided not to make them an additional category of gender given the potential drawbacks of adding unnecessary complexity to the analysis and reducing statistical power. The University of British Columbia Behavioural Research Ethics Board provided approval for this study (approval number: H18-02724).

### 2.2. Variables and Measurements

#### 2.2.1. Outcomes

Two variables were created to indicate the number of unique forms of workplace violence experienced by nurses and instigated by: (1) patients or their visitors and (2) organizational employees, including physicians, nursing co-workers, allied health providers, and management. For each source, participants were asked if they had experienced any of these five forms of workplace violence over the last year: physical assault, threat of assault, emotional abuse, verbal sexual harassment, or sexual assault. Based on participants’ responses, workplace violence was quantified to a numeric value ranging from zero (i.e., having experienced none of the forms workplace violence) to five (i.e., having experienced all five forms of workplace violence) for both instigator sources (i.e., patients or their visitors and organizational employees).

#### 2.2.2. Predictors

Predictors in this study were workplace conditions measured by the 13-factor Guarding Minds at Work (GM@W) Survey [24], which is a validated operationalization of the Standard. Each factor is comprised of five items about a specific aspect of nurses’ work environments, for a total of 65 items in the survey. Participants were asked to rate their agreement to each item using a four-point response scale from strongly disagree (1) to strongly agree (4). Previous literature evaluated the internal structure of the GM@W Survey among BC nurses and confirmed the 13-factor structure [17]. In the current study, workplace conditions were reflected by composite factor scores obtained from confirmatory factor analysis (CFA) [25,26], with higher scores representing healthier work environments. The use of factor scores can alleviate multicollinearity issues between predictors and lead to higher accuracies in prediction [27]. The full list of factors and their definitions can be found in Appendix A.

#### 2.2.3. Controls

Five demographic variables were controlled for in this study, including age, gender (i.e., male or female), nursing experience (in years), sector of work (i.e., acute care, community care, or long-term care), and geographical region (i.e., urban, suburban, or rural). Healthcare sector was dummy-coded as sec1 (acute care = 1, long-term care = 0) and sec2 (community care = 1, long-term care = 0), and geographical region as reg1 (urban = 1, rural = 0) and rec2 (suburban = 1, rural = 0).

### 2.3. Data Analysis

Descriptive analysis was conducted to illustrate the demographic characteristics of the study sample. The omega coefficient (ω) was computed for each composite predictor to evaluate internal consistency, with a ω of above 0.8 being good and a ω of 0.7–0.8 being acceptable [28]. A supervised machine learning algorithm, random forest (RF) analysis [29], was used to rank the importance of various work environment factors for outcomes of interest with the R package caret [30]. The 13 GM@W factors were regressed on each of the two workplace violence outcomes, while controlling for age, gender, years of experience, healthcare sector, and geographical region. Compared to the conventional regression models, the RF models that take into account the potential complex interactions between the predictors are shown to have better predictive performance [31]. 

Following other RF studies, a 10-fold cross-validation approach was adopted, and the dataset was split into 70% for training and 30% for testing [32]. Splitting data allows one to examine whether the model that performs well with one dataset (e.g., training data) still has predictive value when used with a new dataset (e.g., testing data). To assess RF modeling performance, two evaluation metrics were used and the root mean square error (RMSE) was calculated for the training and testing sets. Similar RMSE values for both subsets indicate neither overfitting nor underfitting of the data, suggesting good predictability of the RF model. Another evaluation metric was the $R^{2}$, which is the proportion of the outcome variable variation that the model explains. A higher $R^{2}$ represents better model predictive ability. For each predictor, its relative importance was quantified by the average reduction in prediction accuracy caused by its exclusion from the model. The worse prediction a model without a specific predictor gives, the more important that predictor is. Lastly, partial correlations were computed to test the direction of the bivariate association between the GM@W factors and workplace violence outcomes while controlling for demographic variables.

## 3. Results

The majority of the participants were female (*n* = 3726; 91.6%), with a mean age of 40.25 years (SD = 11.52) and mean work experience of 11.03 years (SD = 7.17). In terms of healthcare sector, the proportion of participants who worked in acute care, community care, and long-term care settings were 73.7% (*n* = 2997), 17.4% (*n* = 708), and 8.9% (*n* = 361), respectively. For geographical region, the proportion of participants who worked in urban, suburban, and rural areas were 62.3% (*n* = 2532), 18.0% (*n* = 731), and 19.7% (*n* = 803), respectively.

Descriptive statistics and reliability coefficients of the 13 GM@W factors are presented in Table 1. Table 2 shows the total number of unique forms of workplace violence participants had experienced. Obtained with CFA, using a centering approach, the mean of each predictor was zero. Internal consistency reliability was good for 12 of the work environment factors (McDonald’ ω ≥ 0.8), and acceptable for the psychological job fit factor (McDonald’ ω = 0.74). More than 86% of the participants reported having experienced at least one form of workplace violence from patients or their families/visitors over the last year (Type II). Workplace violence from organizational employees (Type III) was less frequent, with slightly less than half (48%) reporting exposure to at least one type of workplace violence over the last year.

The relative importance of workplace conditions in predicting workplace violence was ranked through RF regressions (results can be found in Table 3). For both outcomes, the training and testing data had similar RMSE values, which implied good prediction ability of the models. Collectively, the predictors accounted for 22% and 14% of the variance for Types II and III workplace violence, respectively. For Type II violence, *physical safety* (importance score: 27.45) was the most important predictor, followed by *workload management* (importance score: 26.44) and *psychological protection* (importance score: 22.56). For Type III violence, the three top-ranked important predictors were *civility and respect* (importance score: 23.06), *organizational culture* (importance score: 19.62), and *psychological support* (importance score: 19.15). The results of partial correlations demonstrated that the 13 GM@W factors were all negatively associated with both workplace violence outcomes. In other words, nurses who reported working in healthier work environments tended to report fewer exposures to both types of workplace violence. 

## 4. Discussion

This study had two key findings. First, the strongest workplace predictors of Type II and III workplace violence committed against nurses were distinct. While lack of *physical safety*, *workload management,* and *psychological protection* were the three most predictive factors of Type II violence, lack of *civility and respect*, *organizational culture*, and *psychological support* were the three top-ranked predictors for Type III violence. Second, the Standard’s psychosocial factors were stronger predictors of Type II than Type III violence. 

The Standard describes a physically safe workplace (importance score: 27.45) as one where employers take appropriate actions to address physical hazards in the workplace through a variety of measures such as training opportunities and access to equipment and tools required to perform work in a physically safe manner [17,18]. Perhaps unsurprisingly, we found that nurses who reported poorer physical safety in their workplace were also more likely to report greater exposure to multiple forms of violence perpetrated by patients and their visitors. This finding is consistent with the findings of a recent qualitative study that reported insufficient security measures, failure to protect nurses, and insufficient infrastructure and equipment as factors that increase the likelihood of exposure to Type II workplace violence [33]. On the contrary, physical safety was the least important predictor of Type III workplace violence, which may be attributed to the infrequent nature of physical forms of Type III workplace violence. Spector and colleagues (2014) found that under 10% of all cases of physical workplace violence towards nurses were instigated by organizational employees, as opposed to patients and their families/friends which accounted for over 90% of incidents of physical violence in the workplace towards nurses [4]. 

Workload management (importance score: 26.44) was also found to be one of the most predictive factors for Type II workplace violence. Workplaces with appropriate workload management are defined in the Standard as workplaces where assigned tasks and responsibilities can be accomplished successfully within the time available. Effective workload management is dependent on a variety of factors, including access to adequate equipment and resources (human and non-human), minimum unnecessary interruptions and disruptions, and a leader open to discussing staffing and workload [17,18]. This finding is consistent with previous research. A recent cross-sectional study in China found that higher workload among public healthcare workers was associated with increased violence from patients and families [34]. Previous studies have suggested that heavy workload increases the likelihood of Type II workplace violence through its impact on quality and safe care delivery [3,12,13,35]. In other words, when nurses are overworked to the point that they are unable to protect their own health and safety (e.g., increased burnout) and that of their patients (e.g., missed care and medication errors), they are more likely to experience exposure to aggression and violence from patients and their families who are concerned about their safety and wellbeing. 

Lastly, psychological protection (importance score: 23.06) was the third most predictive factor of Type II workplace violence. Psychological protection is defined as a work environment where employees’ psychological safety is ensured through minimizing unnecessary stressors, making efforts to prevent harmful situations from unsafe behaviors by clients, and working to prevent harassment, discrimination, and bullying [17,18]. Interestingly, this factor was not among the top predictors of workplace violence from organizational employees; instead, psychological support (importance factor: 19.15) was one of the top three factors associated with Type III violence. Psychological support is defined as a work environment where the organization is supportive of employees’ psychological health concerns and provides assistance and services or benefits to distressed employees as needed [17,18]. The difference in these predictive factors of Type II and III workplace violence may be explained by the difference in the focus on psychological protection versus psychological harm. While psychological protection has a greater emphasis on preventing psychological harm, psychological support focuses on supporting and treating psychologically harmed employees. This may point to greater support being available for nurses that have experienced violence from outsiders (i.e., patients and visitors) compared to those that experience violence from fellow co-workers. Perhaps the culture around supporting victims of Type II workplace violence is much more salient than those that experience Type III violence, and that lack of support further increases the risk of additional Type III violence (e.g., the lack of assistance for victims of bullying, and, relatedly, lack of consequences for perpetrators of bullying, may make increase the likelihood for this violence to occur). 

Other important predictors of Type III workplace violence were civility and respect (importance score: 23.06) and organizational culture (importance score: 19.62). Civility and respect describe when employees are respectful and considerate in their interactions with one another and are characterized by fair treatment of employees and effective conflict resolution, while organizational culture is rooted in trust, honesty, and fairness, and involves employees feeling that they are a part of an inclusive workplace community and that everyone is treated fairly [17,18]. Previous research has consistently linked respectful communication and a positive organizational culture to fewer incidents of workplace bullying [36,37]. 

In addition to the noted differences between the predictors of Type II and III violence, this study found that the Standard’s psychosocial factors are better predictors of Type II than Type III violence. This finding suggests that working conditions may more prominently increase the likelihood of patients and visitors demonstrating aggressive behaviors compared to nurses and other organizational employees. When working conditions are suboptimal to the extent that they compromise nurses’ ability to deliver quality and safe patient care, patients and visitors may be more likely to demonstrate frustration and aggression towards healthcare workers who are perceived as the root cause of their health and safety concerns [12]. On the contrary, and due to their training, healthcare workers may reason that shortcomings in work environments are not the fault of their co-workers, but rather a result of a multitude of systemic and structural constraints in our healthcare policy and design. Healthcare workers are further bound to their professional code of ethics and standards of practice, even in the face of suboptimal working conditions [38,39]. These different understandings might explain the stronger relationships between working conditions and Type II violence as opposed to Type III. 

### 4.1. Strengths and Limitations

In order to identify the relative importance of 13 different psychosocial factors in predicting workplace violence, this study utilized a machine learning technique (i.e., random forest regression analysis), which is generally regarded as a more appropriate approach compared to linear regression analysis [40,41]. Furthermore, the GM@W survey is a comprehensive and validated tool that operationalizes the Standard, allowing for a thorough and replicable investigation of the relationship between working conditions and workplace violence. Despite these strengths, our study had several limitations. First, our study did not differentiate between the various factors for different forms of workplace violence (e.g., predictors of physical violence may be quite different from predictors of psychological violence). Secondly, the data for this study were collected through surveys, which introduced a potential for recall bias. Additionally, caution should be taken when generalizing our findings beyond the study sample (however, it should be noted that a descriptive comparison of demographics between our sample with the provincial nursing workforce demonstrated less than a 10% difference; blinded for review). Finally, cause-and-effect conclusions cannot be made due to the cross-sectional nature of the study. Additional research should be conducted to determine whether a causal relationship exists between the 13 psychosocial factors outlined in the Standard and Type II and III workplace violence in nursing environments. 

### 4.2. Implications 

The aim of this study was to identify which of the 13 psychosocial risk factors, as outlined in the Standard, are most predictive of Type II and III workplace violence. This study has generated novel findings that provide some direction for practice and research. Most importantly, we found that Type II violence was more prominent (reported by more than 86% of participants) than Type III violence (reported by 48% of participants), and that distinct workplace conditions were stronger predictors of each type of workplace violence. The high prevalence of Type II violence, in various forms, is a serious source of concern and suggests an urgent need for better policies and intervention that more effectively support prevention and management of this violence perpetrated by patients and their visitors. 

Although our findings were generated from a cross-sectional study and cause and effect relationships between factors cannot be assumed, they provide some direction for leaders and practitioners in terms of working conditions that may require further investment and intervention in order to prevent Type II and III violence against nurses. We found that physical safety, workload management, and psychological protection were among the top predictors of Type II violence; in contrast, the strongest predictors of Type III violence were civility and respect, organizational culture, and psychological support. The National Standard website provides a wide range of evidence-based strategies and resources for leaders and employers to address each of the psychosocial risk factors in the workplace [42]. As an example, recommended strategies for ensuring psychological protection in the workplace include, but are not limited to, offering programs and services for those in vulnerable work situations, such as hot zones of workplace violence; these programs and resources may include debriefing sessions, peer support, safe walk program, and secure parking access. Another recommendation is for leaders to be well-educated about their organization’s policies and programs regarding conflict resolution, workplace harassment, discrimination and violence. While reviewing all of the Standard’s recommended strategies and resources is beyond the scope and the space of this paper, the key takeaway is that predictors of Type II and III workplace violence vary. Future research needs to more carefully examine the relationship between these working conditions and various types and sources of workplace violence using more advanced research designs, such as longitudinal studies. In the meantime, however, we have some generic recommendations for leaders and practitioners including creating easy pathways to assess and address workplace violence and its psychosocial factors in the workplace, engaging in training opportunities to better prevent and manage workplace violence, and, perhaps most essentially, fostering dialogue to better support nurses from exposure to various forms and types of WPV. A study by Havaei and colleagues (2019) has shown that nurses who reported that their employers listened to them had higher perceptions of safety [43]. The inclusion of nurses’ voices in developing strategies and policies to reduce violence in their workplaces is of utmost importance. 

The findings in our study provide some direction for practitioners, educators, and leaders in terms of workplace areas requiring investment and intervention to address Type II versus Type III workplace violence and point to the need to assess and address each type of violence individually. In order to prevent Type II violence, which the vast majority of our study participants reported having experienced within the last year, investments in security (e.g., security personnel) and workload management systems and strategies may be a successful approach. On the other hand, investing in shifting internal workplace culture would likely be a more effective strategy for reducing and mitigating Type III violence for nurses. 

## 5. Conclusions

This study identified the top working conditions, as specified in the Standard, associated with workplace violence instigated by patients or their visitors (Type II violence) and organizational employees (Type III violence). Results indicated that Type II violence was much more prevalent than Type III violence, with nearly 87% of nurse respondents indicating they had experienced workplace violence perpetrated by patients or their visitors, whereas less than half of the respondents indicated that they had experienced violence instigated by fellow employees. Of particular note, the top three factors related to Type II violence (namely, lack of physical safety, workload management, and psychological protection) were distinct from the top three factors related to Type III violence (namely, lack of civility and respect, organizational culture, and psychological support). These findings highlight that different psychosocial factors in the workplace may be antecedents to Type II and III violence outcomes. Future research should utilize the Standard to determine if a causal relationship exists between psychosocial factors and incidents of workplace violence committed against nurses, and to determine how each factor is related to various types of workplace violence (i.e., physical assault, threat of assault, emotional abuse, verbal sexual harassment, and sexual assault). 

## Figures and Tables

**Table 1 healthcare-11-01008-t001:** Descriptive statistics and reliability coefficients of the 13 GM@W factors (*n* = 4066).

	Mean	SD	Min	Max	McDonald’ ω
1. Psychological support	0	0.49	−1.16	1.16	0.83
2. Organizational culture	0	0.56	−1.38	1.43	0.81
3. Leadership expectations	0	0.28	−0.69	0.67	0.83
4. Civility and respect	0	0.59	−1.49	1.43	0.83
5. Psychological job fit	0	0.58	−1.76	1.44	0.74
6. Growth and development	0	0.61	−1.65	1.41	0.82
7. Recognition and reward	0	0.55	−1.21	1.33	0.84
8. Involvement and influence	0	0.59	−1.52	1.39	0.82
9. Workload management	0	0.66	−1.58	1.71	0.81
10. Engagement	0	0.46	−2.18	0.68	0.80
11. Balance	0	0.65	−1.55	1.61	0.81
12. Psychological protection	0	0.64	−1.39	1.61	0.87
13. Physical safety	0	0.75	−1.63	1.56	0.89

**Table 2 healthcare-11-01008-t002:** Descriptive statistics of nurse exposure to total number of unique forms of Type II and III violence (*n* = 4066).

Total Number of Unique Forms of Violence	Type II Violence		Type III Violence	
	*n*	%	*n*	%
0	540	13.3	2100	51.6
1	400	9.8	1511	37.2
2	604	14.9	376	9.2
3	963	23.7	60	1.5
4	1193	29.3	16	0.4
5	366	9.0	3	0.1

Note. Type II violence refers to workplace violence instigated by patients and families. Type III violence refers to workplace violence instigated by organizational employees.

**Table 3 healthcare-11-01008-t003:** Workplace conditions regressed on two workplace violence outcomes using random forest regression analyses.

	Type II Violence (-)	Type III Violence (-)
1. Psychological support	16.73	19.15
2. Organizational culture	18.99	19.62
3. Leadership expectations	19.62	18.69
4. Civility and respect	17.95	23.06
5. Psychological job fit	18.98	16.35
6. Growth and development	15.58	13.43
7. Recognition and reward	17.13	14.21
8. Involvement and influence	18.89	13.45
9. Workload management	26.44	16.51
10. Engagement	13.18	11.60
11. Balance	18.32	16.05
12. Psychological protection	22.56	15.78
13. Physical safety	27.45	11.10
$R^{2}$	0.22	0.14
RMSE (train)	1.37	0.71
RMSE (test)	1.38	0.70

Note. RMSE = root mean square error. Control variables (i.e., age, gender, experience, healthcare sector, and geographical region) were included in the models. The negative signs (-) represents the direction of the relationship between the predictors and outcomes. Type II violence refers to workplace violence instigated by patients and families. Type III violence refers to workplace violence instigated by organizational employees.

## Data Availability

Data are available upon reasonable request and approval from the Research Ethics Boards of the University of British Columbia and the British Columbia Nurses’ Union.

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
