# Peer review of "Workplace Predictors of Violence against Nurses Using Machine Learning Techniques: A Cross-Sectional Study Utilizing the National Standard of Psychological Workplace Health and Safety"

_healthcare, 2023, doi:10.3390/healthcare11071008_

Round 1
Reviewer 1 Report
A really excellent article with my full recommendation for publication. It details a real breakthrough piece of research with some interesting findings for those interested in workplace violence - the novelty of this research is that it has been able to identify the different working conditions that are associated with workplace violence across two dimensions (Type II by patients or their visitors and Type III organizational employees). There is good analysis and discussion of the results and the paper is very well presented and argued.
A small suggestion is that it might be worth referring to psychosocial risks given that these are now included in the recently ratified ILO Convention No. 190 in Canada, which covers all forms of violence and harasssment, puts the issue in an OSH context and addresses risks including risks from third parties, risk related to power inequalities and psychosocial risks. If this information is available it would also be interesting to know if there were any conclusions relating to gender of victims / aggressors, or if power / hierarchy had a role to play in relation to organisational violence.
Author Response
Dear Reviewer,
|
Reviewer 1 |
|
|
A really excellent article with my full recommendation for publication. It details a real breakthrough piece of research with some interesting findings for those interested in workplace violence - the novelty of this research is that it has been able to identify the different working conditions that are associated with workplace violence across two dimensions (Type II by patients or their visitors and Type III organizational employees). There is good analysis and discussion of the results and the paper is very well presented and argued. A small suggestion is that it might be worth referring to psychosocial risks given that these are now included in the recently ratified ILO Convention No. 190 in Canada, which covers all forms of violence and harasssment, puts the issue in an OSH context and addresses risks including risks from third parties, risk related to power inequalities and psychosocial risks. If this information is available it would also be interesting to know if there were any conclusions relating to gender of victims / aggressors, or if power / hierarchy had a role to play in relation to organisational violence. |
We really appreciate your positive evaluation of our manuscript and agree with you about the importance and the timeliness of the research.
The term “workplace risk factors” was changed to psychosocial factors or risk factors throughout the paper
While we agree with you that examining the gender and hierarchy of the victims and aggressors is an interesting area for exploration, this information was not sought in our survey and subsequently we are unable to conduct the proposed analysis.
|
For more details please see the revised version manuscript.
Reviewer 2 Report
Thank you for this submission- workplace violence against nurses is a significant issue.
1. The purpose statement needs to be past tense (you state the purpose IS when it should be WAS since it is completed).
2. Why did you remove participants who did not report their gender? Need to clarify.
3. Was a theoretical framework used to guide this study? Most nursing research is guided by theory.
Author Response
Dear Reviewer,
|
Reviewer 2 |
|
|
1. The purpose statement needs to be past tense (you state the purpose IS when it should be WAS since it is completed). |
Thank you for noting this issue. This statement has been removed from the manuscript (since we already have a purpose statement included elsewhere in the manuscript). |
|
2. Why did you remove participants who did not report their gender? Need to clarify. |
Responses: We appreciate the reviewer's comment and would like to provide further clarification on why we excluded participants who did not report their gender. While an additional category could have been created for these missing values, the number of participants in this category was quite small (less than 2% of the total sample), which would not significantly impact our conclusions. Furthermore, including an additional category for these participants may have added unnecessary complexity to the analysis and may have reduced the statistical power of our study. Hence, participants did not report their gender were excluded from the analysis.
Actions: page 3 |
|
3. Was a theoretical framework used to guide this study? Most nursing research is guided by theory. |
We added a paragraph to explain that this study was informed by the validated nursing work life model [lines 109-115] |
For more details please see the revised version manuscript.
Reviewer 3 Report
Dear aurthors
Your manuscript presents an interesting study that will make a significant contribution to the field of nursing as well as healthcare administrations, ether there are a lot of limitations reported.
The study has been carried out by particularly interesting methodology.
I suggest you to remove from lines 41 to 42 the purpose of the study, because you report it in lines 105-107.
I wish you all the best of success
Author Response
Dear Reviewer,
|
Reviewer 3 |
|
|
Your manuscript presents an interesting study that will make a significant contribution to the field of nursing as well as healthcare administrations, ether there are a lot of limitations reported. The study has been carried out by particularly interesting methodology. I suggest you to remove from lines 41 to 42 the purpose of the study, because you report it in lines 105-107. |
We appreciate this positive comment. We have been very particularly with clearly outlining the strengths and the limitations of our study because we believe the study findings should be carefully interpreted in the context of these strengths and limitations. As per your recommendation, lines 41-42 have been removed. |
For more details please see the revised version manuscript.
Reviewer 4 Report
First of all, the topic is very important and sensitive, and similar research can be applied to other categories of employees in healthcare institutions, as well as to other activities where any type of violence is present (with minor adjustments).
Regarding the data collection and the sample, can you explain if it is the same research that you did in the paper: Workplace Predictors of Quality and Safe Patient Care Delivery Among Nurses Using Machine Learning Techniques? The sample size is the same, the time of the survey is also the same, but the response rate is different.
I'm really not sure that this paper has any significant scientific contribution, given that you applied already existing predictors that are defined in the standard and measured their significance from a nursing perspective. Also, you haven't given any specific recommendations on how to overcome those problems. I think that the emphasis of this work should be on that, because just ranking the predictors and classifying them by types of violence does not seem enough to me. Researching your scientific work, it is noticeable that this research has already been published a lot from different aspects, but the fact that each of those papers had some original scientific contribution that you very nicely highlighted in those papers, which is certainly missing here.
I will not delve into further reviewing the paper, because I really think that the direction and essence of the work should be changed. You have a good basis, there is a lack of recommendations or some more serious conclusions, which can perhaps be drawn from the association with demographic variables, especially emphasizing the response between men and women, as well as age. Exactly what you have written somewhere for future research directions, you should add in this paper so that it has some contribution. A lot of similarities with the work I mentioned at the beginning. I am of the opinion that you need to go a step further in this paper.
Author Response
Dear Reviewer,
|
Reviewer 4 |
|
|
First of all, the topic is very important and sensitive, and similar research can be applied to other categories of employees in healthcare institutions, as well as to other activities where any type of violence is present (with minor adjustments). |
Thank you for this positive comment. |
|
Regarding the data collection and the sample, can you explain if it is the same research that you did in the paper: Workplace Predictors of Quality and Safe Patient Care Delivery Among Nurses Using Machine Learning Techniques? The sample size is the same, the time of the survey is also the same, but the response rate is different. |
Thank you for noting this data entry error, which has now been fixed in the manuscript. The data set used in this study is identical to the other two studies of the National Standard cited here, and therefore the response rate is 12%. |
|
I'm really not sure that this paper has any significant scientific contribution, given that you applied already existing predictors that are defined in the standard and measured their significance from a nursing perspective. Also, you haven't given any specific recommendations on how to overcome those problems. I think that the emphasis of this work should be on that, because just ranking the predictors and classifying them by types of violence does not seem enough to me. Researching your scientific work, it is noticeable that this research has already been published a lot from different aspects, but the fact that each of those papers had some original scientific contribution that you very nicely highlighted in those papers, which is certainly missing here.
I will not delve into further reviewing the paper, because I really think that the direction and essence of the work should be changed. You have a good basis, there is a lack of recommendations or some more serious conclusions, which can perhaps be drawn from the association with demographic variables, especially emphasizing the response between men and women, as well as age. Exactly what you have written somewhere for future research directions, you should add in this paper so that it has some contribution. A lot of similarities with the work I mentioned at the beginning. I am of the opinion that you need to go a step further in this paper. |
While we appreciate this comment, we respectfully disagree with the reviewers’ comment regarding lack or limited scientific contribution of this paper. This research study has generated previously undocumented evidence pertinent to the relationship between factors outlined in the National Standards and workplace violence from various instigators. Previous research in this area has two key limitations: (1) failing to operationalize working conditions using measures as comprehensive as the National Standard and/or (2) failing to differentiate between various types of WPV, namely Type II and III, in relation to factors in the workplace. Our study addresses both of these limitations.
Furthermore, the existing research with the National Standard looks at completely different outcomes, mental health symptoms and quality and safe patient care delivery. Notably, the findings of these studies are different and provide direction to decision makers in relation to strategies and interventions for specific problems (e.g., WPV versus high rates of burnout) in the workplace.
Our study recommendation is focused on raising the awareness of decision makers regarding the difference in factors predicting Type II versus Type III WPV against nurses. For example, a leader in an environment struggling with Type II versus Type III WPV would want to look into different aspects of their work environment, as supported by our research findings.
The National Standard provides many evidence-based resources and strategies for employers and leaders to address specific factors in the workplace. While specifying all of these resources is beyond the scope of this paper, we have added a few statements speaking to their availability [see line 354-364] |
For more details please see the revised version manuscript.
Round 2
Reviewer 4 Report
Thank you for explanation and comments.